# The Impact of Post-Pancreatectomy Acute Pancreatitis (PPAP) on Long-Term Outcomes after Pancreaticoduodenectomy: A Single-Center Propensity-Score-Matched Analysis According to the International Study Group of Pancreatic Surgery (ISGPS) Definition

**DOI:** 10.3390/cancers15102691

**Published:** 2023-05-10

**Authors:** Giuseppe Quero, Claudio Fiorillo, Giuseppe Massimiani, Chiara Lucinato, Roberta Menghi, Fabio Longo, Vito Laterza, Carlo Alberto Schena, Davide De Sio, Fausto Rosa, Valerio Papa, Antonio Pio Tortorelli, Vincenzo Tondolo, Sergio Alfieri

**Affiliations:** 1Gemelli Pancreatic Center, CRMPG (Advanced Pancreatic Research Center), Fondazione Policlinico Universitario “Agostino Gemelli” IRCCS, Largo Agostino Gemelli, 8, 00168 Rome, Italy; claudio.fiorillo@policlinicogemelli.it (C.F.); giuseppe.massimiani01@icatt.it (G.M.); chiara.lucinato01@icatt.it (C.L.); roberta.menghi@policlinicogemelli.it (R.M.); fabio.longo@guest.policlinicogemelli.it (F.L.); vito.laterza01@icatt.it (V.L.); carloalberto.schena01@icatt.it (C.A.S.); davide.desio@guest.policlinicogemelli.it (D.D.S.); fausto.rosa@policlinicogemelli.it (F.R.); valerio.papa@policlinicogemelli.it (V.P.); antoniopio.tortorelli@policlinicogemelli.it (A.P.T.); sergio.alfieri@policlinicogemelli.it (S.A.); 2General Surgery Residency Program, Università Cattolica del Sacro Cuore di Roma, Largo Francesco Vito 1, 00168 Rome, Italy; 3General Surgery Unit, Fatebenefratelli Isola Tiberina–Gemelli Isola, Via di Ponte Quattro Capi, 39, 00186 Rome, Italy; vincenzo.tondolo@policlinicogemelli.it

**Keywords:** pancreaticoduodeneectomy, post-pancreatectomy acute pancreatitis, long-term outcomes

## Abstract

**Simple Summary:**

This study investigates the potential impact of post-pancreatectomy acute pancreatitis (PPAP) on long-term outcomes after pancreaticoduodenectomy (PD). Patients who underwent PD from 2006 to 2021 were enrolled in the study. Thirty-two patients developed PPAP and were matched to 32 patients who did not present PPAP post-operatively. PPAP onset was related to a worse post-operative clinical course. No difference was evidenced in terms of overall survival between groups. However, although not statistically significant, patients with PPAP had worse disease-free survival as compared to the no-PPAP cohort.

**Abstract:**

Post-pancreatectomy acute pancreatitis (PPAP) is a potentially life-threating complication. Although multiple authors demonstrated PPAP as a predisposing feature for a more detrimental clinical course, no evidence is currently present on its potential impact on long-term outcomes. The aim of this study is to evaluate how PPAP onset may influence overall (OS) and disease-free survival (DSF) after pancreaticoduodenectomy (PD) for pancreatic ductal adenocarcinoma (PDAC). Patients who underwent PD for PDAC from 2006 to 2021 were enrolled. PPAP was defined according to the International Study Group of Pancreatic Surgery (ISGPS) definition. Propensity score matching (PSM) was performed in order to reduce potential selection biases. After PSM, 32 patients out of 231 PDs who developed PPAP (PPAP group) were matched to 32 patients who did not present PPAP (no-PPAP group). PPAP patients more frequently presented major post-operative complications (*p* = 0.02) and post-operative pancreatic fistula (POPF) (*p* = 0.003). Median follow-up was 26.2 months, with no difference between the two groups (*p* = 0.79). A comparable rate of local or distant metastases was noted in the two cohorts (*p* = 0.2). Five-year OS was comparable between the two populations (39.3% and 35.7% for the no-PPAP and PPAP populations, respectively; *p* = 0.53). Conversely, despite not being statistically significant, a worse 5-year DFS was evidenced in the case of PPAP (23.2%) as compared to the absence of PPAP (37.4%) (*p* = 0.51). With the limitations due to the small sample size, PPAP may potentially relate to worse long-term outcomes in terms of DFS. However, further studies with wider study populations are still needed in order to better clarify the prognostic role of PPAP.

## 1. Introduction

Post-pancreatectomy acute pancreatitis (PPAP) is currently recognized as a rare but potentially severe complication [1,2,3,4,5,6]. Nevertheless, until the recent introduction of the novel grading system of PPAP by the International Study Group of Pancreatic Surgery (ISGPS) [7], no standard definition was present in the literature. This revealed heterogeneous data in terms of causative factors, prevalence and clinical consequences of PPAP on the post-operative course [8,9,10].

The recent ISGPS definition based the diagnosis and grading of PPAP on clinical, radiological, and biochemical features [7]. For instance, the evidence of the sole increase in the serum amylase values, namely post-operative hyperamylasemia (POH), is classified as an independent clinical entity to be distinct from grade B and C PPAP, which imply a deviation from the normal clinical course. The introduction of this novel grading system was aimed to standardize the definition of PPAP in order to specifically assess the impact of PPAP onset on the post-operative course and guarantee (after opportune validation) adequate diagnostic and treatment criteria in the near future.

In this regard, preliminary reports have validated the clinical relevance of the ISGPS PPAP classification, reporting a statistically significant association between PPAP and a more detrimental clinical course [11,12,13]. For instance, PPAP was associated with a higher rate of post-operative Clavien–Dindo ≥3 complications, post-operative pancreatic fistula (POPF), delayed gastric emptying (DGE) and post-pancreatectomy hemorrhage (PPH). On the contrary, no evidence is currently present on the potential impact that PPAP may have on long-term outcomes after pancreatic resection for pancreatic ductal adenocarcinoma (PDAC), namely overall survival (OS) and disease-free survival (DFS). Indeed, an inflammatory microenvironment is widely recognized as a key component in tumor growth and metastasis development considering the complex interaction between cancer cells, immune cells, inflammatory cells, and stromal elements [14].

In this context, the novel and objective PPAP definition and grading may provide significant support in determining the potential association between local inflammation and long-term outcomes.

The aim of this study is thus to apply the ISGPS PPAP classification to a retrospective cohort of patients who underwent pancreaticoduodenectomy (PD) for PDAC in a tertiary referral center in order to define the potential influence of post-operative local pancreatic inflammation on OS and DFS. In order to accomplish this purpose and reduce potential selection biases, a propensity-score-matching analysis (PSM) was conducted on the study population.

## 2. Materials and Methods

### 2.1. Patient Selection and Data Collection

After Institution Review Board (IRB) approval, all patients who underwent PD at the Pancreatic Surgery Unit of the Fondazione Policlinico “Agostino Gemelli” IRCCS of Rome for a histologically proven diagnosis of PDAC from 2006 to 2021 were retrospectively enrolled in the study.

Patients were subsequently divided according to the onset of PPAP. More specifically, patients who developed PPAP constituted the PPAP group, while those who did not present PPAP represented the control group (no-PPAP group). Patients who underwent neoadjuvant treatment (NAT) were excluded from the analysis for the proven influence of NAT on long-term outcomes [15]. Since patients with PPAP were significantly fewer in comparison to the no-PPAP cohort, a PSM was performed in order to minimize potential biases between the two populations.

Perioperative data were retrospectively collected from a prospectively maintained database. Clinico-demographic characteristics included sex, age, body mass index (BMI), and American Society of Anesthesiologists (ASA) score. The following intra- and post-operative features were also collected: operative time, estimated blood loss (EBL), pancreatic texture (firm or soft), Wirsung duct diameter (≤3 mm or >3 mm), associated vascular resection, post-operative serum amylase values from post-operative day (POD) 1 to 3, post-operative complications, and length of hospital stay (LOS).

Clavien–Dindo classification was used to classify post-operative complications [16], while POPF, DGE, and PPH were defined and classified according to the ISGPS criteria [17,18,19]. Post-operative mortality was defined as any death occurring within 30 days from surgery.

In addition, the following histopathological data were analyzed: tumor dimension and grading, number of harvested lymph nodes, and number of positive lymph nodes. The 8th edition of the AJCC/UICC system was used for TNM staging [20].

Long-term outcomes evaluated were OS and DFS. OS was defined as the time between PD and last follow-up or death, while DFS was defined as the time between surgery and the diagnosis of local or distant tumor recurrence.

### 2.2. PPAP Definition and Grading

The ISGPS definition and severity grading system were used for PPAP classification [7]. Specifically, the sustained increase of the serum amylase value over the institutional upper limit for at least 48 h without any deviation from the conventional clinical course was defined as POH. Since no clinical consequences are present in cases of POH, PPAP should not be reported.

The concomitant evidence of a sustained increase of serum amylase values for more than 48 h from surgery, together with a change in patients’ clinical condition and radiological evidence of PPAP requiring pharmacological therapy or endoscopic/interventional radiology procedures, was classified as grade B PPAP.

Lastly, grade C PPAP was defined as the worsening of grade B leading to single or multiple organ failure for at least 48 h, reoperation, or death.

According to the serum amylase upper limit of our institution, POH was defined as the elevation of serum amylase over 100 U/L for at least 48 h from PD.

All CT scans performed were retrospectively revised by dedicated radiologists for the study’s purposes.

### 2.3. Surgical Procedure and Serum Amylase Value Analysis

As previously reported [21,22,23], a Whipple procedure with Child’s reconstruction was performed in all cases. Specifically, all patients underwent a duct-to-mucosa pancreaticojejunostomy with internal stent positioning. The same bowel loop was then used for the hepaticojejunostomy and gastrojejunostomy. The gastojejunal anastomosis was performed at least 60 cm from the hepaticojejunostomy in a side-to-side, antecolic, and antiperistaltic manner.

According to the internal institutional protocol, serum amylase values were routinely evaluated on POD 1 and 3 and more rarely on POD 2. Consequently, in case of absence of POD 2 values, POH was defined as a sustained increase of serum amylase values on POD1 and POD 3. Neither steroids nor somatostatin analogs were used in the post-operative period.

### 2.4. Study Outcomes

The primary endpoint of the study was to compare the PPAP and no-PPAP cohorts in terms of DFS and tumor recurrence. Secondary endpoint were an additional comparison between the two study populations in terms of OS and post-operative clinical outcomes.

### 2.5. Statistical Analysis

The effect of confounding factors and selection bias was reduced by calculating the propensity score using logistic regression. The PSM analysis was conducted for the following potential confounding factors: age, sex, ASA score tumor grading, and TNM staging. Patients were, thus, matched on these propensity scores at a 1:1 ratio. An optimal matching with a caliper size of 0.2 was used to avoid poor matches.

Continuous data were reported as median and quartile rank (QR), while all categorical variables were expressed as number and percentages. Student’s *t*-tests, Mann–Whitney U tests, Fisher’s tests, and χ^2^ tests were used for the univariate analysis. Kaplan–Meier log-rank survival analysis was performed to evaluate the correlation of each clinicopathologic feature with OS and DFS. Uni- and multivariate regression analyses were performed using Cox proportional hazards models, and hazard ratios (HRs) are reported with 95% confidence intervals (CIs)

A *p* value < 0.05 was considered statistically significant. All tests were performed using SPSS version 25 for Windows (SPSS Inc., Chicago, IL, USA).

## 3. Results

From January 2006 to December 2021, 478 patients underwent PD at the Fondazione Policlinico Universitario “Agostino Gemelli” IRCCS of Rome. In total, 266 of them (55.6%) presented a histologically proven diagnosis of PDAC and thus represented the study cohort. Before PSM, 29 patients (10.9%) who underwent NAT and 6 patients (2.3%) lost at the follow-up were excluded. Of the remaining 231 patients, 32 (13.8%) developed PPAP, while POH was evidenced in 28 cases (12.1%). After PSM, 32 patients who presented PPAP (PPAP group) were matched to 32 patients who did not develop PPAP (no-PPAP group) (Figure 1).

All patients in the PPAP group underwent a CT scan post-operatively. No statically significant difference was noted between the two study groups in terms of clinico-demographic characteristics, namely age, sex, ASA score, BMI and preoperative diagnosis of diabetes (Table 1).

### 3.1. Intra- and Post-Operative Outcomes (Table 1)

The two study groups presented comparable operative time (*p* = 0.39) and EBL values (*p* = 0.31), while patients who developed PPAP had a higher rate of soft pancreatic texture (23–71.9% vs. 15–46.9% in the no-PPAP cohort; *p* = 0.04) and a pancreatic duct ≤3 mm (24–75% vs. 16–50% in the no-PPAP group; *p* = 0.04). As expected, post-operative serum amylase values were significantly higher in the PPAP group compared to the no-PPAP population both in POD 1 (*p* < 0.0001) and POD 2-3 (*p* < 0.0001). In terms of post-operative course, PPAP patients more frequently presented Clavien–Dindo III–IV complications (21–65.6% vs. 9–28.1%; *p* = 0.002). In particular, PPAP onset was related to a higher incidence of POPF (28–87.5% vs. 17–53.1%; *p* = 0.003) and led to a more severe manifestation of it (14–43.8% grade B and C) as compared to the no-PPAP group (6–18.7% grade B and C) (*p* = 0.01). Although only approaching significance, DGE and PPH presented a higher frequency in the PPAP population than in the no-PPAP group (*p* = 0.06 and 0.08, respectively). An equal rate of in-hospital mortality was evidenced in the two cohorts, while hospitalization was significantly longer in the PPAP group than the no-PPAP cohort (19 (12–24) days and 13 (11–18) days, respectively; *p* = 0.01).

In terms of histopathological findings (Table 2), the two study populations presented similar tumor dimensions (*p* = 0.63), number of harvested lymph nodes (*p* = 0.1), and number of positive lymph nodes (*p* = 0.91). All patients presented negative resection margins.

### 3.2. Long-Term Outcomes and Multivariate Analysis of Prognostic Factors for OS and DFS

Follow-up was completed in all patients, with median values of 33 (12.6–1181.1) months and 24.4 (17.9–76.5) months for the no-PPAP and PPAP groups, respectively (*p* = 0.79). As a whole, adjuvant therapy was prescribed to 199 out of 231 patients (86.1%). However, only 67.8% of them (135 patients) finally received adjuvant chemotherapy. The main reasons for exclusion from treatment were poor general clinical conditions (25–12.6%), severe post-operative complications (21–10.6%), personal patients’ reasons (7–3.5%), and patients lost at follow-up (11–5.5%). With regards to our study populations, no difference was documented between the two cohorts in terms of adjuvant chemotherapy (12 patients (37.5%) and 17 patients 53.1%) for the PPAP and no-PPAP groups, respectively; *p* = 0.81). Moreover, no statistically significant difference was evidenced in terms of median time elapse between surgery and access to adjuvant treatment (7.1 vs. 5.8 weeks for PPAP and no-PPAP patients; *p* = 0.12). Tumor recurrence was similarly evidenced in the two study populations (11 cases (34.4%) and 16 cases (50%) in the no-PPAP and PPAP groups, respectively; *p* = 0.2) with no difference in terms of site of recurrence (*p* = 0.97).

Notably, the 5-year OS did not differ between the two cohorts (39.3% and 35.7% in the no-PPAP and PPAP groups, respectively; *p* = 0.53). Similarly, no difference was evidenced in terms of DFS, although a lower value was reported for the PPAP population (23.2%) in comparison to the no-PPAP cohort (37.4%) (*p* = 0.51) (Figure 2a,b).

An analysis of the potential prognostic factors influencing OS and DFS was additionally conducted (Table 3). In the univariate analysis, OS was significantly influenced by stage III tumors (*p* = 0.01), tumor grading 3 (*p* = 0.04), and the presence of lymph node metastases (*p* = 0.001). All these features were confirmed as influencing factors on 5-year OS at the multivariate analysis.

With regards to the 5-year DFS, stage III tumors (*p* = 0.001), tumor grading 3 (*p* = 0.04), and the presence of metastatic lymph nodes (*p* < 0.0001) were the only variables significantly associated with a worse DFS in the univariate analysis. Of them, only stage III tumors (HR: 5.14, 95% CI: 1.06–24.8; *p* = 0.04) and positive lymph nodes (HR: 2.69, 95% CI: 1.6–11.69; *p* = 0.03) were recognized as influencing factors on the 5-year DFS in the multivariate analysis.

## 4. Discussion

The onset of acute pancreatitis after pancreatic resection is currently recognized as a frightening post-operative complication, especially for its known correlation with the potential development of local and systemic adverse events [1,4,5,6,11,12,13]. However, until the recent introduction of the ISGPS criteria [7], no uniform definition of PPAP was present in the literature. The majority of authors based the definition of PPAP on the Atlanta criteria [24], while others used the biochemical alteration of post-operative serum amylase levels alone as diagnostic tool [25]. This inevitably revealed heterogeneous data in terms of PPAP incidence rate and discrepancies in the potential influence of PPAP on post-operative outcomes. In this context, the novel definition criteria proposed by the ISGPS [7] were precisely conceived to provide a unanimous definition of PPAP in order to objectively and uniformly assess the effective correlation between PPAP and patients’ outcomes both in terms of post-operative clinical course and prognosis.

However, although several authors already confirmed the ISGPS grading as a valuable tool in defining the impact of PPAP on short-term outcomes [11,12,13], no evidence is currently present on the potential influencing role that PPAP may have on OS and DFS. This may find justification not only in the absence of a uniform definition of PPAP until recently but also in the rare incidence rate of clinically significant PPAP described so far, which makes study cohorts too small to draw solid conclusions. Given these premises, the main aim of this study was to evaluate, for the first time in the literature, the potential influence that PPAP may have on patients’ prognosis after PD for PDAC using the novel and objective classification of the ISGPS.

According to our results, no difference was evidenced between the no-PPAP and PPAP populations in terms of OS, with a 5-year value of 39.3% and 35.7%, respectively (*p* = 0.53). Similarly, no statically significant difference was evidenced between the two cohorts in terms of 5-year DFS, although a worse outcome was noted for the PPAP group (23.2%) as compared to the no-PPAP population (37.4%) (*p* = 0.51). This difference between the two cohorts (although not significant) should not be underestimated, especially in relation to the small sample size of our study cohort.

Indeed, the potential correlation between inflammation and tumor development and progression attracted the attention of the scientific community, especially in the last decades [14,26,27,28]. However, although several authors demonstrated how the preoperative onset of acute pancreatitis negatively affects long-term outcomes after pancreatic resection, evidence on the potential role that PPAP may have on patients’ prognoses is still lacking.

For instance, preoperative acute pancreatitis has been demonstrated to be significantly related to lower survival and higher risk of tumor recurrence after pancreatic resection [29,30]. Specifically, Feng et al. [31] recognized acute pancreatitis as a negative prognostic factor for early recurrence, with a 3.57-fold higher risk compared to patients without a preoperative pancreatic acute inflammation. Despite this evidence, the underlying mechanism of correlation between preoperative acute pancreatitis and patients’ prognosis is still unclear. Two main hypotheses have been proposed. First, the onset of acute pancreatitis may delay surgical treatment and make patients more prone to the development of post-operative complications, thus affecting the therapeutic role of surgery [32,33]. The second hypothesis is based on the physiopathological process that originates from inflammation. More precisely, the onset of local inflammation facilitates vascular permeability, leading to the invasion of blood and lymphatic vessels by cancer cells, which may cause the subsequent onset of local or distant metastases [14].

Based on this last postulation, it is likely that a similar event may also occur in case of post-operative development of acute pancreatitis. Indeed, the presence of local and systemic inflammation have been recognized as negative prognostic factors for survival after the treatment of multiple malignancies, increasing the local and distance recurrence rate at the same time [34,35]. In this process, the physiological release of proinflammatory cytokines, here including interleukin (IL)-1, IL-6, IL-8, and tumor necrosis factor alpha (TNF-a), seem to play a key role [36,37]. In particular, the increased levels of IL-6 have the capability to decrease the number and development of T lymphocytes, which—along with the concomitant immunosuppression due to post-surgical stress response and general anesthesia—would facilitate the growth of occult micrometastases [38]. Although not in a statistically significant way, the hypothesis proposed may justify the discrepancy in terms of DFS between the no-PPAP and PPAP cohorts of our study population.

The significant limitation represented by the small sample size of our study population is furthermore supported by the results obtained when post-operative complications, and specifically POPF, were analyzed as potential influencing features on long-term outcomes. As matter of fact, several authors recognized both the post-operative development of severe complications and POPF as related to a worse prognosis [39,40]. Van der Gaag et al. [39] reported a 2.06-fold risk of dismal prognosis in case of major surgical complications. Similarly, Nagai et al. [40] documented a higher risk of peritoneal recurrence in case of clinically significant POPF, with a hazard ratio value of 3.974. Overlooking our data, although not in a statistically significant way, the post-operative onset of complications presented a tendency towards worse long-term outcomes, with a 5-year DFS of 22.6% in case of Clavien–Dindo≥3 complications as compared to Clavien–Dindo grade 1–2 (54%). Similar results have also been found when comparing clinically significant POPF (5-year DFS: 11.2%) to the cohort of patients who did not develop POPF (39.5%).

With regards to the short-term outcomes, we already reported a strong correlation between the development of PPAP and a more severe clinical course. Specifically, the onset of grade B and C PPAP was associated with a higher rate of severe post-operative complications, DGE, POPF, PPH, and reoperation [11,12,13]. These findings were confirmed in the current cohort of study, further supporting the results already reported by Chen et al. [13] and Ikenaga et al. [12]. Even for the short-term outcomes, the underlying physiopathological process still needs to be elucidated. It is likely that the onset of local inflammation may lead to a local and systemic response, causing more frequently severe and potentially life-threatening complications.

Our study presents several limitations. Firstly, the small sample size of the study population represents the main drawback and does not permit us to draw solid conclusions in spite of a tendency towards a worse long-term clinical course in the case of PPAP onset. Secondly, the retrospective study design conducted in a prolonged time-elapse may represent a significant limitation for the generalization of the results. On the contrary, for the first time in the literature, we demonstrated the potential influence that PPAP may have on DFS after pancreatic resection for PDAC. This finding is furthermore strengthened by the PSM conducted on the study population, including only patients affected by PDAC in the analysis.

## 5. Conclusions

In conclusion, despite the onset of PPAP having been confirmed as a significant risk factor for a worse post-operative clinical course and perhaps being a feature that could potentially influence long-term outcomes (especially in terms of DFS), the need for further studies on larger cohorts of patients in order to objectively assess the impact that PPAP may have on oncological outcomes after pancreatic resection for PDAC is undeniable.

## Figures and Tables

**Figure 1 cancers-15-02691-f001:**
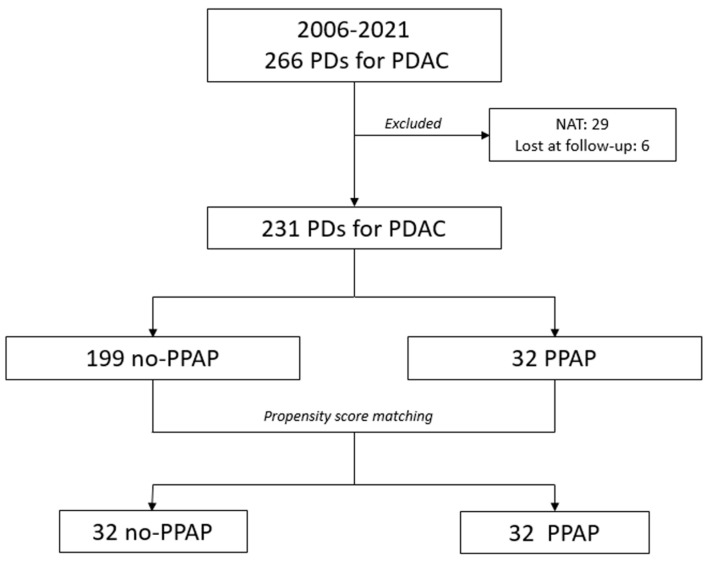
Flowchart showing the distribution of patients in two groups before and after propensity score matching (PSM).

**Figure 2 cancers-15-02691-f002:**
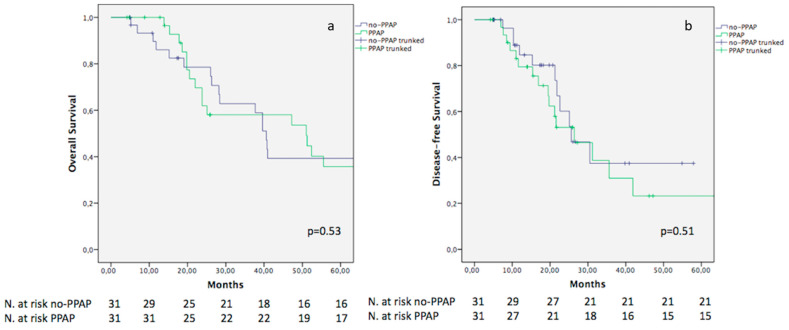
Kaplan–Meier curves comparing OS (**a**) and DFS (**b**) between no-PPAP and PPAP.

**Table 1 cancers-15-02691-t001:** Clinico-demographic and perioperative characteristics of the no-PPAP and PPAP cohorts.

	No-PPAP (n = 32)	PPAP (n = 32)	*p*
Preoperative data			
Sex ratio (M:F)	0.78	1	0.61
Age (years), median (QR)	69.5 (58.2–74.7)	70 (58.2–75.7)	0.98
ASA score, n (%)			
I	2 (6.3)	2 (6.3)	0.5
II	24 (75)	20 (62.5)	
III	6 (18.7)	10 (31.2)	
BMI, median (QR)	24.4 (21.3–26.3)	25.5 (20.9–27.6)	0.85
Diabetes, n (%)	4 (12.5)	6 (18.8)	0.49
Intra-operative outcomes			
Operative time (min), median (QR)	360 (312–375)	362.5 (315–415)	0.39
EBL (mL), median (QR)	221.7 (188–269)	245.4 (201–296)	0.31
Pancreatic texture, n (%)			
Firm	17 (53.1)	9 (28.1)	0.04
Soft	15 (46.9)	23 (71.9)	
Pancreatic duct, n (%)			
≤3 mm	16 (50)	24 (75)	0.04
>3 mm	16 (50)	8 (25)	
Vascular resection, n (%)	7 (21.9)	9 (28.1)	0.56
Post-operative outcomes			
Serum amylase (U/l), median (QR)			
POD 1	37 (14.2–79.2)	411.5 (277.5–1115.2)	<0.0001
POD 2-3	53.5 (32.2–83.5)	476.5 (349–678)	<0.0001
Post-operative complications, n (%)			0.002
Clavien–Dindo I–II	17 (53.1)	9 (28.1)	
Clavien–Dindo III–IV	9 (28.1)	21 (65.6)	
PPAP grade, n (%)			
POH	4	-	
B	-	30	
C	-	2	
POPF, n (%)	17 (53.1)	28 (87.5)	0.003
POPF grade, n (%)			
BL	11 (34.4)	14 (43.8)	0.01
B	4 (12.5)	6 (18.8)	
C	2 (6.3)	8 (25)	
DGE, n (%)	8 (25)	15 (46.9)	0.06
PPH, n (%)	0	3 (9.4)	0.08
Reoperation, n (%)	6 (18.8)	9 (28.1)	0.37
LOS (days), median (QR)	13 (11–18)	19 (12–24)	0.01
In-hospital mortality, n (%)	1 (3.1)	1 (3.1)	1

ASA: American Society of Anesthesiologists; BMI: body mass index; EBL: estimated blood loss; PPAP: post-pancreatectomy acute pancreatitis; POPF: post-operative pancreatic fistula; DGE: delayed gastric emptying; PPH: post-pancreatectomy hemorrhage; LOS: length of hospital stay.

**Table 2 cancers-15-02691-t002:** Histopathological features and follow-up outcomes.

	No-PPAP (n = 32)	PPAP (n = 32)	*p*
Tumor dimension (cm), median (QR)	2.19 (1.85–2.98)	2.21 (1.77–3.01)	0.63
TNM staging, n (%)			
I	18 (56.2)	11 (34.4)	0.21
II	11 (34.4)	16 (50)	
III	3 (9.4)	5 (15.6)	
Harvested lymph nodes, median (QR)	18.5 (14–26)	22.5 (16–27)	0.1
Positive lymph nodes, median (QR)	2 (1–2.5)	2 (1–2.5)	0.91
Tumor grading, n (%)			
G1	5 (15.6)	6 (18.8)	0.67
G2	24 (75)	21 (65.6)	
G3	3 (9.4)	5 (15.6)	
Follow up (months), median (QR)	33 (12.6–118.1)	24.4 (17.9–76.5)	0.79
Adjuvant chemotherapy, n (%)	12 (37.5)	17 (53.1)	0.81
Recurrence, n (%)	11 (34.4)	16 (50)	0.2
Site of recurrence, n (%)			
Local	2 (6.3)	3 (9.4)	0.97
Distant	9 (28.1)	13 (40.6)	

**Table 3 cancers-15-02691-t003:** Univariate and multivariate analysis for OS and DFS.

Variable	Univariate Analysis				Multivariate Analysis					
	5-Year OS (%)	*p*	5-Year DFS (%)	*p*	5-Year OS			5-Year DFS		
					HR	95% CI	*p*	HR	95% CI	*p*
Age ≤ 67/>67	25.6/44.2	0.06	32.1/25	0.4	-	-	-	-	-	-
Sex, M/F	34.1/40.4	0.19	19.5/43.3	0.3	-	-	-	-	-	-
ASA, I-II/III	33.1/50.3	0.17	33.1/38.6	0.5	-	-	-	-	-	-
BMI, ≤24/>24	24.2/47.3	0.17	23.9/33.9	0.09						
EBL, ≤225/>225 Ml *	31.8/42.3	0.78	30.9/25.2	0.42	-	-	-	-	-	-
Clavien-Dindo, I-II/III-IV	42.9/38.9	0.69	54/22.6	0.33	-	-	-	-	-	-
PPAP, yes/no	35.7/39.3	0.53	23.2/37.4	0.51	-	-	-	-	-	-
POPF, B-C/no	21.5/43.3	0.84	11.2/39.5	0.21	-	-	-	-	-	-
Reoperation, yes/no	30.8/39.5	0.81	21.5/32.8	0.97	-	-	-	-	-	-
TNM, I/II-III	53.3/21.3	0.01	78.7/11.2	0.001	14.9	1.99–31.9	0.01	5.14	1.06–24.8	0.04
Tumor grading, 1–2/3	72/30.1	0.04	66.7/23.6	0.04	2.91	1.45–8.08	0.05	4.06	0.4–41.52	0.23
Lymph nodes harvested, ≤21/>21	49/30.8	0.49	38.1/14.2	0.31	-	-	-	-	-	-
Positive lymph nodes, yes/no	58.7/15.5	0.001	59.1/10.6	<0.0001	4.33	1.03–18.1	0.04	2.69	1.6–11.69	0.03

* The mean values were used as the cutoff for the univariate and multivariate analyses. ASA: American Society of Anesthesiologists; BMI: body mass index; EBL: estimated blood loss; PPAP: post-pancreatectomy acute pancreatitis; POPF: post-operative pancreatic fistula.

## Data Availability

Data are available from the corresponding author upon reasonable request.

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
