# Peer review of "The Impact of Post-Pancreatectomy Acute Pancreatitis (PPAP) on Long-Term Outcomes after Pancreaticoduodenectomy: A Single-Center Propensity-Score-Matched Analysis According to the International Study Group of Pancreatic Surgery (ISGPS) Definition"

_cancers, 2023, doi:10.3390/cancers15102691_

Round 1
Reviewer 1 Report
The study is not hypothesis driven.
1) Too many abbreviations, makes it easier to write but harder to read. 2) Postpancreaticoduodenectomy pancreatitis was defined as as an increased amylsae level. A lot of difference between pancreatitis and increased amylase levels. 3) The writing is poor, (e.g., "no-PPAP group). 4} patients without PPAP had firmer glands and smaller pancreatic ducts, fewer postoperative serious complications, fewer POPF (which seem very frequent in general). Length of stay is long. Adjuvant therapy was infrequently applied. Survival is not bad, without differences between groups. Thhis can be accepted with a rewrite if it is in the best interests of the journal, but iin its own merit it is of low to moderate priority.
Respectfully,
nothing to add.
Reviewer 2 Report
The manuscript analyses the impact of post-pancreatectomy acute pancreatitis (PPAP) on long-term outcomes after pancreaticoduodenectomy. 32 patients (out of 231) were included in this single center retrospective series, and matched to 32 controls. Survival was not significantly different between groups.
This is a potentially interesting analysis. There are, however, several points the authors might want to address:
Why was the matching done in a 1:1 fashion, and not (especially considering the small cohort) in a 1:2 or 1:3 fashion? This would potentially increase the power of the analysis
It is quite surprising and counter-intuitive that the rate of adjuvant therapy was (although not significant) higher in the PPAP groups (53 vs 38%). Do the authors have any explanation for this? Could the authors provide data on the rate of adjuvant therapy in the whole cohort (and maybe also in those patients without complications.
Would it be possible to analyse grade B and C PPAP separately?
none - only minor checks required.
